# Tissue-Engineered Nanomaterials Play Diverse Roles in Bone Injury Repair

**DOI:** 10.3390/nano13091449

**Published:** 2023-04-24

**Authors:** Teng Wan, Meng Zhang, Hao-Ran Jiang, Yi-Chong Zhang, Xiao-Meng Zhang, Yi-Lin Wang, Pei-Xun Zhang

**Affiliations:** 1Department of Orthopedics and Trauma, Peking University People’s Hospital, Beijing 100044, China; 2Key Laboratory of Trauma and Neural Regeneration, Peking University, Beijing 100044, China; 3National Center for Trauma Medicine, Beijing 100044, China

**Keywords:** nanomaterial, bone tissue injury, engineered bone tissue, nanoparticle, drug loading, targeted therapy

## Abstract

Nanomaterials with bone-mimicking characteristics and easily internalized by the cell could create suitable microenvironments in which to regulate the therapeutic effects of bone regeneration. This review provides an overview of the current state-of-the-art research in developing and using nanomaterials for better bone injury repair. First, an overview of the hierarchical architecture from the macroscale to the nanoscale of natural bone is presented, as these bone tissue microstructures and compositions are the basis for constructing bone substitutes. Next, urgent clinical issues associated with bone injury that require resolution and the potential of nanomaterials to overcome them are discussed. Finally, nanomaterials are classified as inorganic or organic based on their chemical properties. Their basic characteristics and the results of related bone engineering studies are described. This review describes theoretical and technical bases for the development of innovative methods for repairing damaged bone and should inspire therapeutic strategies with potential for clinical applications.

## 1. Introduction

Bone is a hierarchically sophisticated composite nanomaterial that plays significant roles in humans by supporting key structures, maintaining movement stability, and stabilising the internal environment. It is an adaptive and dynamic system comprising inorganic and organic materials with a precise structure from nanoscale to macroscale [1]. Bone is structurally rigid and elastic, but it can be compromised due to tumours, trauma, infection, and aging. Bone repair is a complex and continuous process that involves restoring the integrity of bone tissues, including structural and functional reconstruction.

Stable incomplete fractures can self-heal without specific treatment because bone tissues can be completely auto-regenerated and reconstructed into original tissues. Complete fractures can heal after surgical reduction and firm fixation [2]. However, self-healing only works within a specific size range; thus, large defects beyond a critical size and bone tissue reconstruction cannot be completed [3]. In addition to conventional reduction and fixation, bone autografts and allografts are necessary for repairing large, critical bone defects. Autografts are regarded as the current “gold standard” for healing such defects, as they can support structural stability and provide growth factors, pre-osteoblasts, and extracellular matrix for bone regeneration. However, they can be limited by contagion, infection, inflammation, and limited donor organisation [4]. Thus, considerable effort has been invested in developing innovative treatment strategies to promote the formation of new bone and enhance the effects of therapies to help heal damaged bone.

Recent efforts have been directed towards developing advanced biomaterials, such as engineered bone tissues and nanomaterials that promote bone regrowth at damaged sites, then clinically applying them [5]. Nanomaterials have at least one dimension of a three-dimensional spatial scale from 1–100 nm [6]. They have specific physical, chemical, and biological properties, such as electrical conductivity, superparamagnetism, high surface area-to-volume ratios, and high surface energy. They can be designed for drug loading, fluorescent labelling, targeted therapy, and imaging diagnosis [7]. The structure of nanomaterials is similar to that of natural bone, and they can pass through cell membranes and directly activate stem cell bioactivity. Nanomaterials can also indirectly regulate osteoblast-related cell proliferation, differentiation, and migration by being incorporated into composite scaffolds to enhance their hydrophilicity, electrical conductivity, and mechanical energy [8].

As shown in Figure 1, this article reviews and discusses the applications of nanomaterials as well as strategies for fabricating bone regeneration materials. The macro- and nano-hierarchical architecture of normal bone has been systematically determined (Figure 1a). Nano-based materials for bone tissue engineering can achieve rapid bone regeneration by their nano effect, structure bionics, composition simulation, and targeted delivery, which are needed for clinical applications (Figure 1b). Nanomaterials are classified as inorganic or organic based on their chemical composition (Figure 1c). The properties, merits, applications, and fabrication strategies with promising results for representative bone engineering nanomaterials are also explained (Figure 1d).

## 2. Macro- and Nano-Hierarchical Architecture of Natural Bone

Natural bone is a heterogeneous, anisotropic, organised nanocomposite material, the principal components of which are hierarchically organised from the macro- to the nanoscale [9] (Figure 2). Studies have struggled with the biomimetic architecture of natural bone tissue. Autologous bone grafting is currently the gold standard for bone regeneration, and nothing has yet surpassed it.

### 2.1. Macroscopic Architecture

Overall, the human skeletal system comprises trabecular and cortical bones in a ratio of ~4:1. However, the ratios differ according to functional and stress requirements [10].

#### 2.1.1. Cortical Bone

Cortical (compact) bone is mainly distributed in the long diaphysis and the surfaces of various other types of bone, and it is characterised by dense, hard, and strong pressure resistance. Compact bone forms circumferential lamellae on the outer and medullary cavity surfaces of the diaphysis, and Haversian lamellae are formed in the middle of it.

Outer circumferential lamellae are located parallel to the superficial area of the diaphysis. The periosteal covers the surface of the outer circumferential lamellae and branches out from blood vessels deep into bone tissue via the Volkmann canal [11]. The inner circumferential lamellae are located in the lateral bone marrow cavity of the diaphysis. They also have a perforated canal through which small vessels in it communicate with those in bone marrow [12]. The Haversian lamella (Figure 2a) is situated between the inner and outer circumferential lamellae and is arranged in concentric circles centred on a Haversian canal that together form Haversian systems. Haversian canals contain blood vessels, nerves, and some connective tissue. Haversian systems are cylindrical, about 4 mm long and 300 μm wide at their base, and they form branching networks within cortical bone. As the main component of compact bone in the diaphysis, Haversian systems are also called osteons [13].

#### 2.1.2. Trabecular Bone

Spongy, porous, hard and soft tissues mainly comprise trabecular bone that is encased in cortical bone. Trabeculae are needle-like or irregularly thin, rod-shaped layers of parallel bone plates (Figure 2b) [10]. The texture of trabecular tissue is that of a bone meshwork with many interconnecting spaces containing marrow, nerves, and blood vessels [14]. They are mostly distributed in weight-bearing areas, specifically at both ends of long bones and inside vertebrae, as well as short, oblate, and irregular bones.

### 2.2. Nano Architecture

#### 2.2.1. Bone Biomineral Matrix

Bone matrix (Figure 2c) consists of organic and inorganic components, of which the former is mainly secreted by osteoblasts. Nearly 90% of the extracellular organic materials are collagenous fibres (Figure 2d), whereas the remaining 10% is an amorphous matrix mainly composed of glycoprotein complexes with a few sialic acid proteins and lipids [15]. Inorganic components (commonly referred to as bone salts) include hydroxyapatite (HAp), amorphous colloidal calcium phosphate, magnesium, sodium, potassium, and trace elements. Hydroxyapatite is the main inorganic component of human bones; it can promote the repair of defective bone and has biological activity. Hydroxyapatite crystals are needle-shaped or cylinders with a length and width of 20–40 and 2–3 nm, respectively (Figure 2e) [16]. Figure 2c shows HAp crystals aligned parallel to the long axis of collagen fibrils and embedded in gaps between collagen molecules [17]. Changes in the spatial orientation of HAp crystal structures and the relative spatial position of HAp and collagen fibrils cause mineralisation of bone matrix abnormalities.

Organic and inorganic bone biomineral-matrix nanomaterials are considered ideal for processing into bone regeneration scaffolds because they are nontoxic and biocompatible. These natural biomineral-matrix nanomaterials are used to develop a new kind of artificial bone substitute, mimicking natural human bone tissue in both composition and microstructure and applied to repair the segmental defect. For example, nano-HAp has been blended into scaffolds that bridge bone defects because it has an excellent ability to promote osteogenesis and is found in natural bone tissues [18]. Collagen has been incorporated into composites or made into nanoparticles or nanofibres for engineered bone tissue because it can significantly enhance cell adhesion, spreading, migration, and osteogenic activity [19].

#### 2.2.2. Bone Cells

Four main types of bone cells function in the physiological process of fracture healing: osteoblasts, osteocytes, osteoclasts, and extracellular lining cells (Figure 2f). The synergistic action of these cells regulates the absorption and formation of old and new bone tissues, respectively. These bone cells have been extensively exploited as models for engineered bone tissues in vitro. They have been used to verify the biological functions of bone-repair composites and to regulate the balance of physiological target cells in vivo to rapidly repair injured or disordered bone [20].

Anabolic osteoblasts originate from bone marrow mesenchymal stem cells (BMSCs) and are the most widely used for engineering bone tissues in vitro [21]. Osteoblasts synthesise, calcify, and generate new bone at sites of active bone formation due to fracture, poroma, tumours, or infections. Osteoblasts are the primary functional cells involved in bone formation and are responsible for the synthesis, secretion, and mineralisation of the bone matrix. Most bone engineering efforts have aimed to improve bone regeneration at implantation sites by upregulating osteoblast adhesion, proliferation, and secretion of the bone extracellular matrix. Osteoblasts initially synthesise bone matrix by secreting type Ⅰ collagen, osteocalcin (OCN), osteopontin (OPN), osteonectin, and proteoglycans [22]; they then promote the transformation of osteoblast progenitor cells into mature osteoblasts by expressing alkaline phosphatase, osterix (OSX), OCN, and bone sialoprotein (BSP Ⅰ) [23]. This is followed by the secretion of insulin-like growth factor-1 (IGF-1), fibroblast growth factors (FGF), interleukin-1 (IL-1), prostaglandin, etc., to regulate osteogenic activity [24].

Osteocytes account for 90–95% of the total number of cells distributed in the cortical bone mineralised matrix. They are differentiated from osteoblasts. Osteoblasts have three destinies at the final stage of bone formation, which is to differentiate into osteocytes, migrate to the bone surface to become inactive, or enter the programmed death pathway (apoptosis) [21]. Osteocytes are associated with other bone tissue cells, such as osteoblasts and osteoclasts on the bone surface, and with bone marrow cells via dendrites. This allows them to not only send signals to cells at the bone surface but also to affect the proliferation and differentiation of stem cells in the bone marrow [25]. Osteocytes are believed to regulate bone reconstruction by generating osteoclasts and osteoclast signals in response to mechanical stress and transmitting them to osteoblasts and osteoclasts on the bone surface via dendrites [26]. Assessing molecular signal communication and stress responses between osteocytes and bone implants will be indispensable for future bone tissue engineering because it will directly affect the success of internal fixation.

Bone lining cells are quiescent flat osteoblasts that cover the bone surface and are likely to be involved in normal bone remodelling, and they play important roles in the therapeutic responses to anabolic osteoporosis and in the repair of damaged bone [27]. Bone lining cells are about 12 μm in diameter and 1 μm thick, and their processes are connected with those of surrounding bone lining cells and osteocytes in adjacent bone via interstitial junctions. Bone lining cells might transform into active osteoblasts or osteoblast progenitor cells. Evidence indicates that bone lining cells have osteogenic functions in bone formation upon stimulation [28]. Moreover, bone lining cells, together with other types of bone cells, are important components of the bone-metabolising unit, an anatomical structure that exists during bone remodelling cycles [29]. Bone lining cells cooperate with osteoblasts to participate in osteoid mineralisation. One of the most important functions of bone lining cells is the formation of a periosteal barrier that maintains the static state of the bone surface.

Osteoblasts are hematopoietic progenitor cells derived from a mononuclear macrophage cell line. They are the only cells with bone resorption function. The relative balance between osteoclast bone resorption and osteoblast bone formation plays an important role in maintaining bone homeostasis. Under the regulation of osteoblasts, mononuclear osteoclast progeny cells gradually fuse with polynucleated mature osteoclasts, secrete acid substances and protease, dissolve minerals, and digest collagen fibres of bone matrix for bone resorption [30]. At first glance, osteoclasts might damage osseointegration because their overactivation results in osteoporosis and hypo-osteogenesis. In fact, the bone-resorbing action of osteoclasts determines the longevity of bone scaffolds.

## 3. Required Clinical Features of Nanoscaled Materials

Bone can be damaged by almost any pathological insult, such as infection, direct trauma, tumours, or long-term stress. Bone tissue can be completely regenerated under normal healing conditions [31]. However, nonunion and delayed union fractures remain challenges in clinical practice. Segmental defects > 2 cm or 50% of the circumference are unlikely to spontaneously heal after internal fixation of a bone injury because of the absence of a template for regeneration [32]. People with congenital or acquired diseases, such as diabetes, metabolic diseases, infection, neoplasm, and osteoporosis, have an extremely high risk of nonunion fractures. When faced with these issues, the present gold standard of autologous bone grafts has never been surpassed, despite its limitations [33]. Therefore, safe and effective treatment strategies for damaged bone are urgently needed that are not limited to preclinical studies.

The recent advent of nanomaterials has become a potential breakthrough in therapeutic strategies that promote bone regeneration. Nanomaterials are attractive in terms of bone tissue reconstruction because they can precisely mimic the hierarchical and nanoscale features of bones and can be internalised into cell membranes to achieve precise targeted therapy [34]. For example, nanofibres can mimic the structure of the extracellular matrix (ECM) and upregulate the bioactivity of cells associated with bone repair. The macro- and nano-architectures endow bone tissue with unique properties, such as stiffness and toughness, etc., enabling the bone to support the load and thus providing stability to the body. Natural bone tissue has certain piezoelectric properties. Under physiological stress, the bone will generate natural bioelectricity and form natural bone bioelectric signals. This piezoelectric response behavior is a necessary condition for bone growth, remodeling, and regeneration. Nanomaterials with conductivity, piezoelectricity, and reinforcement have been incorporated into traditional bone-repair materials to mimic the mechanical properties of natural bone tissue and regulate bioelectrical signals in bone tissue. All these features facilitate osseointegration at the bone implant/tissue interface [35]. Therefore, using nanomaterials for regenerative medicine might shed light on how to repair damaged bone.

## 4. Nanomaterials in Bone Tissue Engineering

This section discusses the classification and applications of nanomaterials for bone regeneration. The properties and basic characteristics of promising inorganic and organic nanomaterials and recent advances in bone tissue engineering applications are systematically reviewed (Figure 3).

### 4.1. Inorganic Nanomaterials

#### 4.1.1. Metallic Nanoparticles

Versatile metallic nanoparticles (NPs) have been widely applied to tissue engineering, clinical diagnosis, biological imaging, and targeted therapy. This is due to their excellent physicochemical properties, such as biocompatibility, electron storage capacity, controllable shape and size, and high-energy atoms located on their surface [36]. The surfaces of metallic NPs can be modified using functional groups that provide bonding sites for antibodies, ligands, genes, or bioactive factors [37]. Therefore, the surface modification or incorporation of metallic NPs into traditional bone-repair materials can significantly improve their physicochemical and biological properties [38]. Gold (Au) and silver (Ag) NPs have attracted interest for bone tissue engineering and have been applied in clinical practice.

Gold NPs can function as vehicles for delivering all sizes of therapeutic agents and biomolecules, such as proteins, peptides, growth factors, and DNA. Kim et al. [39] developed a gene delivery system with a three-layered putamen structure using AuNPs with different diameters to promote bone regeneration and osteogenic differentiation. Functional gene fragments of activating transcription factor (ATF) 4, osterix (transcription factor SP7), and runt-related transcription factor 2 (RUNX2) combined with AuNPs improved osteogenesis efficiency via cell internalisation. These results showed that loaded AuNPs delivered therapeutic genes that were internalised by human MSCs. Owing to their high X-ray absorption coefficients, good stability, and biocompatibility, AuNPs have been broadly used to construct molecular imaging probes and diagnostic nanoprobes. Mastrogiacomo et al. [40] labelled calcium phosphate bone cement with Au and PLGA nanoparticles to track the progress of bone repair. These are ideal materials for preventing and treating bone infections, owing to excellent conductivity and broad-spectrum bacteriostasis. Positively charged AgNPs can anchor to the walls of negatively charged microbial cells. Silver NPs then pierce cell walls, damage DNA, and inhibit protein production, thus causing the cells to die, as they cannot metabolise or reproduce [41]. Abdelaziz et al. [42] added AuNPs to polycaprolactone (PCL) and polylactic acid/cellulose acetate (PLA/CA) nanofibre scaffolds to promote bone regeneration and prevent postoperative infections. The results of bacteriostatic experiments showed that AgNPs inhibit bacteria within 40 mm in tissues in vitro. Therefore, AgNP-loaded bone-repair scaffolds promote the proliferation and osteogenic differentiation of MSCs and significantly reduce the risk of infection. Composite scaffolds based on metallic nanomaterials, such as AuNPs, AgNPs, and CuNPs, have been studied for bone repair because they readily conduct electricity by allowing free electrons to move between the atoms and can act as particle reinforcement to enhance mechanical performance. AgNP is one of the most electroconductive and stable materials. Shuai et al. [43] modified AgNPs on polydopamine functioned BaTiO_3_ to increase the conductivity of the polyvinylidene fluoride (PVDF)/barium titanate (BaTiO_3_) composites. Results demonstrated that the piezoelectric properties of the bone-repair scaffold modified with silver nanoparticles were significantly enhanced, with output current and voltage increased by 50% and 40%, respectively, relative to that of pure PVDF/BaTiO_3_. Moreover, in their study, with the increase in the amount of AgNPs, the compressive properties and modulus of the bone-repair scaffold showed an initial increase and then a decrease, indicating that incorporating an appropriate amount of silver nanoparticles can enhance the mechanical performance of the scaffold.

#### 4.1.2. Carbon-Based Nanomaterials

Carbon-based nanomaterials such as carbon nanotubes (CNTs), graphene, reduced graphene oxide (rGO), fullerenes, and nanodiamonds (Figure 4) have excellent mechanical, conductive, and osteoinducible properties [44]. Therefore, they have been widely applied to regenerative medicine and tissue engineering for imaging, biosensors, regenerative medicine, and drug delivery [45]. In particular, the excellent electrical conductivity and mechanical enhancement properties of carbon-based nanomaterials make them ideal candidates for use as secondary structure enhancers for building hybrid materials with metals, ceramics, and polymers and developing scaffolds with optimum properties. Graphene and carbon nanotubes are the most prevalent bone tissue engineering materials with the highest potential for clinical transformation [46].

Graphene is a novel nanomaterial with sp2 hybrid linked carbon atoms tightly packed into a single two-dimensional honeycomb lattice [47]. A series of biomedical graphene materials has been developed and investigated in terms of pharmaceutical agent absorption, cell imaging, and conductive composites [48]. Graphene combined with natural or synthetic biological materials enhances the mechanical strength, hydrophilicity, and mechanical energy of composite scaffolds. Shuai et al. [49] developed a high-performance ceramic akermanite composite that was mechanically reinforced by incorporating graphene and boron nitride nanotubes (BNNTs). The mechanical properties were significantly enhanced in these compared with pure akermanite composites. Moreover, the reinforced scaffolds have favourable biocompatibility and bioactivity for bone regeneration due to good conductivity. Li et al. [50] incorporated polydopamine-mediated graphene oxide (PGO) into conductive alginate/gelatin (AG) scaffolds to promote periodontal bone regeneration in diabetic patients. The addition of 10 wt% of PGO to the bone-repair scaffold can increase the maximum conductivity and compression modulus to 1.6 s/cm and 50 kPa, respectively. In vitro and in vivo studies have shown that the conductive PGO/AG scaffold can regulate immune activity, anti-inflammatory activity, oxygen species (ROS)-scavenging, and induce bone regeneration by mediating glycolysis and the RhoA/ROCK pathway in macrophages. The graphene family of nanomaterials can serve as delivery vehicles for bioactive factors and therapeutic drugs. Studies [51,52,53] found that adsorbed drugs or growth factors loaded on graphene or its derivatives can remarkably improve the proliferation, secretion, and osteodifferentiation of osteoprogenitor cells due to prolonged sustained stimulation. Liu et al. [54] loaded bone morphogenetic protein (BMP)-2 onto poly l-lysine-functionalised graphene oxide (GOPLL) to accelerate bone healing. They found that BMP-2 extended GO-PLL/BMP-2 nanocomposite release to over 14 days, which significantly enhanced osteogenic differentiation and matrix calcification and favourable bone formation ability.

Carbon nanotubes (CNTs) are a class of cylindrical nanomaterials composed of graphene sheets. They comprise single (SWNT)- and multi (MWNT)-walled types of carbon nanotubes according to the number of graphene layers. The SWNTs and MWNTs comprise single and nested sheets of rolled graphene, respectively [55]. Conductive bone-repair materials have been constructed using CNTs because they improve electrochemical and electron conduction connections between associated biomolecules and proteins, thus accelerating osteoblast proliferation and bone formation [56]. Osteoblast function is enhanced by electrical signals [57]. Adding CNTs to traditional scaffold materials can significantly improve their mechanical properties and endow biomaterials with electrical conductivity, which are regarded as bone-repair implant materials with clinical application potential [58]. Wang et al. [59] added worm-like helical CNTs to polylactic acid (PLA) composites to confer toughness and conductivity. Tensile and compressive mechanical properties have been significantly improved in CNT/PLA composites compared with pure PLA.

#### 4.1.3. Silica Nanoparticles

Silica (SiO_2_) NPs have a unique structure with excellent performance. They have a large specific surface area, large pores, and good biocompatibility and biodegradability [60]. Bare or porous Silica NPs are simple to prepare according to applications [61]. Current investigations into the applications of silicon NPs are mainly focused on drug delivery systems, biological imaging, and osteogenic differentiation. The effects of hydrophobic or antitumour drugs can be improved by loading them into efficient delivery systems. Optical materials in mesoporous Silica NPs can be modified for biological imaging [62]; bone tissue regeneration materials can be created to promote osteogenic differentiation for tissue engineering research [63].

Silica NPs can be loaded with bone-repair-active factors such as proteins, peptides, cytokines, and functional gene fragments that promote bone regeneration. Shen et al. [64] prepared mesoporous Silica NPs that served as nanocarriers for loaded basic fibroblast growth factor (bFGF). Nanocomposites of bFGF/Silica NPs promoted cell adhesion, proliferation, and osteodifferentiation in vitro, and Silica NPs promoted new bone formation in animal models of femoral defects in vivo. To achieve ideal osteogenic activity, sustained drug release, or appropriate three-dimensional morphology, Silica NPs have been added to organic/inorganic materials to prepare composite bone-repair scaffolds. Martin-Moldes et al. [65] fabricated biomineralised silk-silica bone regeneration composites to verify the effects of Silica NP size on osteoinduction. Their results indicated that Silica NPs yielded nanoparticles ranging from 200 to 500 nm and enhanced osteoinduction. Others have focused on doping Silica-based nanomaterials with other functional elements to confer unique biological functions. Sutthavas et al. [66] synthesised strontium (Sr)- and calcium (Ca)-loaded mesoporous Silica NPs to induce osteogenic stem cell differentiation. Doping with Sr or encasing calcium phosphate in Silica nanoparticles significantly promotes osteogenic differentiation.

#### 4.1.4. Nano-HAp and Other Bone Mineral Substitutes

Nano-hydroxyapatite (nano-HAp) has similar, but not identical, chemical composition (Ca_10_(PO_4_)_6_(OH)_2_) to human bone minerals. These minerals are mainly thin plate-shaped carbonated calcium-deficient hydroxyapatites with a Ca/P ratio of 1.3, higher than pure nano-HAp (1.67) [67]. This deviation is mainly attributable to the carbonated groups, the substitution of other cations (e.g., Ca^2+^ with Mg^2+^), and protonation of ${\mathrm{PO}}_{4}^{3-}$ in the crystal lattice. Hydroxyapatite and other bone mineral substitutes are used to create prosthetic implants, scaffolds, and artificial bone cement [68,69,70]. Nano-hydroxyapatite comprises only calcium, phosphorus, oxygen, and hydrogen, which are physiologically essential and harmless to humans. Therefore, it is stable, has good biological performance, and does not release ions that might induce toxicity or inflammation. In addition, because nano-HAp has similar chemical composition to human bone, it has outstanding osteogenic features of biocompatibility, bioactivity, bone conductivity, and bone induction. Therefore, nano-HAp implants induce cell growth at sites of damaged human bone and promote extracellular matrix secretion that prompts cell proliferation and osteogenic differentiation [71]. Nano-HAp has been applied as a filler to reinforce polymers and improve the mechanical stability of composites, facilitate cell interactions, and promote cell proliferation, adhesion, and differentiation, including that of stem cells [72,73]. Nano-HAp also facilitates cell–material surface interactions through selective protein adsorption.

Modifying nano-HAp by grafting with a polymer or doping it with other ions can improve the mechanical properties and the osteoinductivity of composites [72,74,75]. Park et al. [76] grafted PLGA onto the surface of polymeric nano-HAp by melt grafting with transesterification to enhance bonding force between them. Tensile mechanical property studies have shown that the amount of PLGA increased by more than double when grafted onto nano-HAp composite scaffolds compared with pure nano-HAp-blended composites. Coating with nano-HAp is the most prevalent technique for improving implant bioactivity and osseointegration. For example, coating Mg alloys with nano-HAp enhances corrosion resistance, biocompatibility, and the bioactivity of Mg alloy substrates. Uddin et al. [77] modified Mg alloy implants with nano-HAp and found that coating bone metal implants with nano-HAp significantly improved corrosion resistance and reduced the immune inflammatory response. Nano-HAp has excellent adsorption capacity. Therefore, it allows the slow and controlled release of bioactive factors or drugs and reduces the degradation cycle and toxic side effects of loaded drugs. Chen et al. [78] loaded alendronate (AL) onto nano-HAp crystals and deposited them onto the surface of an engineering-grade PLA filament (EPLA) microcarrier using biomimetic mineralisation to achieve local targeted drug delivery. Nano-hydroxyapatite can also regulate the immune microenvironment through size effect, thereby indirectly regulating the bone tissue regeneration process. Zhao et al. [79] prepared nano-hydroxyapatite of sizes 400 nm and 200 nm, and in vitro cell experiments showed that 400 nm nano-HAp could not only induce M2-phenotype macrophages (M2) polarization to decrease the production of inflammatory cytokines but could also promote the production of osteogenic factors. The results of analysing drug loading and release in vitro indicated that a strong chemical bonding interaction between alendronate and nano-HAp improved drug loading capacity and sustained more alendronate release compared with pure EPLA microspheres.

#### 4.1.5. Black Phosphorus

Black phosphorus (BP) is an emerging two-dimensional material that has attracted considerable scientific attention since the emergence of a new preparation method and the discovery of its excellent electrical conductivity in 2014 [80]. Black phosphorus consists of a single phosphorus element that is highly homologous to the inorganic components of natural bone. Therefore, BP-based biomaterials have significant advantages for bone injury repair [81]. The advantages of BP in bone tissue engineering are as follows: (1) It can be degraded into nontoxic phosphates after oxidation, which then absorb surrounding free calcium ions and binds to them to form mineralized calcium phosphate that is deposited to promote bone formation in situ and repair; (2) the strong light absorption capacity of BP within the near infra-red (NIR) spectrum confers a stable and safe light-controlled release mode on nanomaterials based on it, and this results in more stable and sustained drug release [82]; and (3) Black phosphorus also has good photothermal conversion ability within the NIR spectrum, and local hyperthermia upregulates alkaline phosphatase and heat shock protein, increases mineralized crystal formation, allows longitudinal and concentric bone growth, and thus accelerates bone repair [83].

Recent composite scaffolds have been created to regenerate bone based on the biomineralisation function of BP nanomaterials. Huang et al. [82] developed a composite bone-repairing hydrogel that internally retained the BP nanosheets to accelerate bone regeneration. During degradation, BP releases phosphorus atoms that bind to endogenous calcium atoms and promotes the osteogenic differentiation of new bone tissue. Black phosphorus nanomaterials can act as drug delivery systems due to their negatively charged, highly specific surface areas. Black phosphorus nanomaterials can be coated with various functional bioactive molecules and drugs [84]. Cheng et al. [85] modified BMP-2 on BP nanosheets then blended them with poly-l-lactic acid (PLLA) to form composite electrospun nanofibrous scaffolds. The loaded BMP-2 released from the fibres promoted the proliferation and differentiation of preosteoblasts. The photothermal and photodynamic therapeutic effects of BP nanomaterials on bone regeneration have recently been investigated [86]. Wang et al. [87] developed a BP/poly (lactic-co-glycolic acid) (PLGA) microsphere NIR light-triggered drug release system based on the photothermal effect of BP nanosheets. The BP/SrCl_2_/PLGA microsphere drug delivery system enables precise control of drug delivery at specific times under infrared light.

Attractively, BP not only has excellent electrical conductivity but also is a piezoelectric material and phosphorus-like natural bone tissue [88]. This means that the bone-repair scaffold based on BP implanted in the bone injury site can directly form a charge after being subjected to physiological stress without direct exogenous electrical stimulation. However, the application of BP in bone injury repair is mostly based on its excellent conductivity. It is believed that bone-repair materials based on the piezoelectric properties of BP will be a very promising research direction in the future because appropriate stress under Wolff’s law can promote bone tissue regeneration.

#### 4.1.6. Magnetic Nanoparticles

Magnetic NPs have emerged as potential tools in bone tissue engineering. They can adjust the cell microenvironment by exploiting the inherent magnetic field and enhancing the osteoinductive, osteoconductive, and angiogenic properties of scaffolds [89]. Maghemite (Fe_2_O_3_) or magnetite (Fe_3_O_4_) are magnetic nanoparticles that are extensively used to engineer tissues [90]. Sufficiently small iron oxide nanoparticles become superparamagnetic. That is, they exhibit magnetism in an external magnetic field and cannot be retained even if the magnetic field is removed [91]. This unique superparamagnetic property permits precise magnetic control and is harnessed for cell induction using the ability of each nanoparticle to generate intrinsic magnetic fields. After internalisation by osteoblast-related cells, superparamagnetic nanoparticles can promote the activation of intracellular pathways that facilitate osteogenesis [92].

Magnetic NPs have been incorporated into conventional organic and inorganic biomaterials, and their intrinsic magnetic fields enhance the bone regeneration capacity of composites. Shuai et al. [92] constructed a magnetic microenvironment and incorporated Fe_3_O_4_ magnetic NPs into PLLA/polyglycolide (PGA) scaffolds to enhance cell viability and promote bone regeneration. The intrinsic magnetic field of a composite scaffold with magnetic nanoparticles can upregulate intracellular biological activities. Magnetothermal therapy is a promising noninvasive way to treat tumour-related bone defects. Li et al. [93] added various concentrations of magnetic Fe_3_O_4_ NPs to hydrogel composite scaffolds to achieve magnetothermal conversion. These magnetic composite scaffolds supported bone mesenchymal stem cell differentiation and had favourable antitumour characteristics. Magnetic nanoimaging technology can detect the pathophysiological process of diseases at the cellular, molecular, or genetic level in vivo and realise the early qualitative and quantitative diagnosis of diseases. [94]. Campodoni et al. [95] combined two magnetic NPs with a bone-repair scaffold to trace the process of bone regeneration in vivo. Radioactive tracers that were analysed and evaluated using SPECT/CT techniques showed that 0.75 wt% magnetic NPs mixed into bone scaffolds was sufficient to label bone tissues with magnetic NPs and monitor bone regeneration in vivo.

#### 4.1.7. Nanoclays

Nanoclays, which are a kind of natural or synthetic nanomaterial, have attracted more and more attention in bone tissue engineering due to their good biocompatibility, swelling, rheology, mechanical properties, and cellular uptake [96]. In terms of chemical composition, nanoclay minerals are composed of alumina, silica and water, iron, magnesium, alkalis, and alkaline earth metals. In physical structure, nanoclay minerals are composed of one or more layers of fine-grained or elongated fibrous-shaped phyllosilicate minerals. Nanoclays can be categorised into three different categories according to the specific arrangement of their tetrahedral (SiO_4_) and octahedral (AlO_6_) units [97]. The types of common nanoclays are summarized in Table 1. Studies [98,99] have shown that nanoclay can significantly promote cell adhesion, proliferation, and differentiation by releasing Mg^2+^ and Ca^2+^, which provides a new strategy and method for regenerative medicine based on seed cells. For the future, nanoclay is a promising orthopedic material because it is easy to be chemically functionalized [100], can be used as an additive material to enforce the mechanical performance of bone implantation [101], and can be used as a drug carrier to upregulate the formation of bone and cartilage by releasing bioactive factors [102].

Nanoclay minerals are often associated with organic materials to be employed in the biomedical and tissue regeneration fields. Nanoclay can interact with polymers through surface electrostatics and hydrogen bonding, thereby improving swelling, rheology, mechanical properties, and cellular uptake of composite materials. To mimic the hardness, mechanical performance, and biofunction of natural bone tissue, nanoclays have been added to natural or synthetic polymers to construct bone-repair scaffolds to obtain artificial bone materials with mechanical properties matching that of bone tissue. Indeed, a large number of in vivo animal studies have shown that this nanoclay/ploymer bone-repair scaffold can achieve rapid bone regeneration. Wu et al. [98] constructed a novel biodegradable composite scaffold by hydroxyethyl cellulose (HEC)/soy protein isolate (SPI)/montmorillonite (MMT) to promote bone tissue regeneration. Through the interfacial interaction between nanoclays and cellulose, the mechanical and physical properties of the nano-bone-repair scaffold were achieved in the ideal range. In addition, during the degradation of the MMT/HEC composite scaffold in vivo, magnesium ions and calcium ions were released, which promoted the proliferation and differentiation of BMSCs. Nanoclays have been extensively applied as growth factors, proteins, and therapeutic drug delivery vehicles due to their strong adsorption capacity and high cation exchange. Halloysite nanotubes (HNTs) have two different charges inside and outside the lumen, enabling them to carry proteins with different isoelectric points. Ji et al. [103] used 3D printing technology to prepare a bone-repair scaffold. This scaffold enables sequential and sustained release of deferoxamine and BMP-2 through HNTs and microcarriers to promote angiogenesis and ossification at the bone defect site. Nanoclays can be chemically modified to provide unique and personalized functions, such as immunomodulation, antibacterial activity, and dispersibility. Fereshteh et al. [104] used chitosan grafting to modify HNTs, and this drug delivery vehicle was used to load icariin to construct a chitosan /HNTs/icariin hydrogel. The modification of HNTs with chitosan increased entrapment efficiency and loading capacity with reduced initial burst release of icariin from the nanotubes.

### 4.2. Organic Nanomaterials

#### 4.2.1. Polymeric Nanoparticles

Polymeric NPs are solid particles of 1–1000 nm prepared from natural or synthetic materials. They are modifiable, biocompatible, minimally immunotoxic, and biodegradable [105]. Polymeric NPs can be fabricated using natural polymer compounds such as chitosan, collagen, cellulose, fibroin, and gelatin and synthetic polymers such as PLGA, PGA, PLA, and PCL. Almost all of these compounds are biodegradable and form water and carbon dioxide after degradation. These polymers are considered safer for nanoparticle construction in bone tissue engineering because they are biocompatible, and their degradation products are nontoxic [106]. It is worth noting that polyester compounds such as PLGA, PGA, PLA, and PCL will form acidic intermediates (such as lactic acid, glycolic acid, or adipic acid, etc.) in the process of biodegradation, which limits their further clinical application. These acidic by-products will reduce the pH of the microenvironment at the implant site and cause inflammation, which ultimately affects the process of bone injury repair [107]. Therapeutic drugs or bioactive factors covalently bound to or encapsulated in polymeric NPs can be slowly and continuously released via dissolution, degradation, or dispersion. Moreover, polymer substrates can control the kinetics of drug release and have a protective characteristic that improves therapeutic substances and prolongs the half-life of drugs and stability in vivo.

Various types of therapeutic agents, such as growth factors, biomolecules, or genes, can be loaded into PLGA NPs. For example, Castillo-Santaella et al. [108] developed BMP-2-loaded PLGA NPs to control BMP-2 release. Kinetic findings in vitro have shown that BMP-2-loaded PLGA PNs can achieve optimal release profiles within the shortest term, without an initial burst of loaded growth factors. The advent of polymeric NPs is linked to their nanoscale dimensions, which facilitates their diffusion through membranes and aids cellular uptake. Based on the above characteristics, polymeric NPs have been applied to fluorescence imaging. This has promising opportunities for accurate and early imaging diagnosis of diseases and the timely treatment of bone-related diseases [109]. Chen et al. [110] examined the effects of polymer NPs of different sizes on NIR-II fluorescence imaging in vivo and their size-dependent distribution in Sprague-Dawley rats. Tissue sections showed that 15 nm particles mainly gathered in endothelial cells of blood vessels in the bone marrow cavity. Their findings provided new techniques and ideas for bone imaging and the treatment of bone-related diseases.

#### 4.2.2. Polymeric Nanofibres

Nanofibres have diameters of 1–1000 nm and have the advantages of good biocompatibility and tensile mechanical properties, pore interconnectivity, high physical adsorption capacity, and gas permeation [111]. Nanofibres have great potential in bone tissue engineering because of the highly biomimetic structure of the tissue extracellular matrix (ECM) that is important in mimicking bone tissue (Figure 5b). The ECM in the physiological setting functions as a support for surrounding cells, supplies growth factors to regulate osteoblast-related cell behaviour and provides a suitable cell microenvironment. Nanofibres can mimic this complex structure when membranes constructed with morphological and viscoelastic characteristics similar to those of the ECM promote the growth and synthesis of different tissues [112]. Such scaffolds must promote nutrition and interactions with the tissue microenvironment, as this can lead to neovascularization. Hence, this is considered one of the main goals of successful implantation.

Polymeric nanofibres containing growth factors, stem cells, and targeted drugs (Figure 5e) have recently been applied as bone regeneration scaffolds to promote the repair of bone injuries. Lian et al. [113] developed a double-layer nanofibre membrane with osteogenic and bacteriostatic properties. A dense bacteriostatic membrane layer consisted of doxycycline hydrate loaded with conventional PLGA nanofibres and a loose layer comprising PLGA/gelatin nanofibres combined with dexamethasone-loaded mesoporous SiNPs to promote osteogenesis. Analyses of osteogenesis in vitro revealed that the dense layer of nanofibre membranes significantly promoted the osteogenic differentiation of rat bone marrow stem cells. The dense layer of nanofibres sustained doxycycline hydrate release and was a potently effective antibacterial agent in vitro. Collagen is an abundant natural polymer that comprises about 90% of the organic materials in the human bone matrix [114]. Collagen fabricated into nanofibres to mimic the mineralised structure of the bone matrix had excellent bioabsorbability and bone conductivity. Gu et al. [115] covalently crosslinked polyphosphate to a collagen nanofibre scaffold to prevent bleeding and alveolar bone resorption after extraction. These collagen nanofibre scaffolds promoted bone regeneration in situ and preserved the alveolar ridge in rat models of alveolar bone defects. Pure nanofibre does not have the function of bone scaffold for supporting and fixation of fracture ends, as it is the membrane structure. However, nanofibres can be used as secondary materials to incorporate into other polymers for enhancing the mechanical performance of composite scaffolds. Pinho et al. [116] prepared the chitosan nanofibres using electrospinning technology, which is applied to enhance the mechanical properties of composite materials. The tensile mechanical properties showed that the tensile modulus of the composites increased by 70% after the introduction of nanofibres.

#### 4.2.3. DNA Nanomaterials

Genetic DNA nanomaterial has been applied to bioimaging, diagnosis, drug delivery, anti-infection, and targeted therapy [117]. Such DNA nanomaterials can now be accurately designed, modified, and endowed with unique functions. These nanomaterials have been developed to functionally change cell biological processes, such as migration, proliferation, differentiation, autophagy, and anti-inflammatory activities [118].

During bone repair, DNA nanomaterials are applied to modify scaffolds that support bone or as a delivery system to overcome the poor penetration and low stability of other biological agents [119]. Targeted therapy with DNA nanomaterials is promising for bone tissue repair, as it can overcome the need for supraphysiological dosages and side effects of recombinant human growth factors, such as heterotopic ossification and soft tissue swelling. Kim et al. [120] found that BMP2-pDNA on bioactive glass NPs promotes the biological activity of MSCs. Targeted-release efficiency was extremely high in vitro, as >70% of bioactive glass NPs loaded with BMP2-pDNA was internalised by cells. These findings suggested that the BMP2-pDNA delivery system is suitable for bone regeneration. Nonviral gene vectors have been developed by combining DNA with other scaffolds to improve therapeutic effects on bone regeneration. Lei et al. [121] developed nonviral gene delivery nanoparticles in various hydrogels to increase the applicability of gene therapy to tissue regeneration and cancer therapy. They found that scaffolds incorporating nanogenes are effective for precise local targeted therapy in vivo.

#### 4.2.4. Liposomes and Exosomes

Liposomes are vesicles composed of at least one spherical lipid bilayer in aqueous solution that structurally resembles the lipid membranes of live cells. The lipid bilayer is mainly composed of amphiphilic compounds such as glyceryl phosphatide and sphingomyelin, or synthetic block copolymers, which form ordered nanostructures in aqueous solution (Figure 6a) [122]. These self-assembled, core-shell vesicle structures enable liposomes to carry hydrophobic and hydrophilic drugs (Figure 6d). Such phospholipid bilayer nanomaterials might be useful tools for manipulating cell and tissue responses to bone-repair scaffolds and achieving targeted delivery through modification with diverse ligands (Figure 6b) [123]. Liposome-loaded composite scaffolds have been designed to exploit the biocompatibility of liposomes and robust scaffolds to provide a novel system suitable for clinical applications. Mohammadi et al. [124] grafted liposomes loaded with BMP-2 peptide onto the surface of nano-HA/PLA nanofibre scaffolds to promote bone regeneration. They found that liposomes are effective platforms for the sustained release of BMP-2 peptides and considerably promote the osteogenic differentiation of MSCs. Therefore, liposomes could be unique candidates for bone regeneration therapeutics.

Exosomes are extracellular vesicles of 40–160 nm in diameter that are secreted by cells and carry biomolecules such as DNA, RNA, lipids, metabolites, and cytoplasmic and cell-surface proteins. They are also responsible for cell–cell signal communications. Figure 6 shows that exosomes with a nanostructure similar to liposomes are also composed of lipid bilayers. They are extracellular vesicles (EVs) secreted by various types of natural cells and are found in physiological fluids such as plasma, interstitial fluid, and saliva [125]. Exosomes in humans transport loaded contents from host to recipient cells, enabling information exchange between them (Figure 6c). Exosomes have excellent biomedical potential as functional nanomaterials, as they can be active agents and delivery systems for advanced drug delivery, tissue engineering, and disease therapy. Ma et al. [126] connected BMSC-derived exosomes through the exosomal capture peptide CRHSQMTVTSRL (CP05) with an injectable thermosensitive hydrogel for bone regeneration. The results showed that the exosomes provided a bionic microenvironment that enhanced the proliferation and osteogenic differentiation of BMSCs and provided superior bone repair.

## 5. Summary and Perspectives

This review discusses nanomaterials as repair materials for bone regeneration based on their basic characteristics, typical applications, and biofabrication strategies. The applications of these nanomaterials to bone tissue engineering have been thoroughly reviewed to help develop promising new approaches to treating damaged bone, along with encouraging results in vitro and in vivo. Applications and the biofabrication of engineered bone mainly include the following strategies: (1) Creating vehicles to deliver controlled-release nanomedicines to targeted lesion sites; (2) incorporating nanoregenerative materials into bone-repair composite scaffolds that mimic the natural bone matrix in terms of composition and microstructure and applying them to repair segmental defects; (3) developing nanodiagnostics by labelling fluorescence probes to achieve rapid diagnoses and early point-of-care tests for diseases; and (4) achieving early diagnosis and therapeutic intervention for bone tumours or osteoporosis, etc., at the cellular and molecular levels using molecular imaging. The unique characteristics of various nanomaterials and their typical applications in bone tissue engineering are summarised in Table 2.

Nanoscale therapeutic materials such as nano-HA and gelatin have been used to prepare composite engineered bone tissues to repair large defects and enhance osseointegration during the internal fixation of fractures in the clinical setting. However, significantly more work is required to evaluate local aggregation, biodegradation, toxicity, and inflammatory responses to nanoscale bone-regeneration material designed for clinical translation. Specifically, more focus on assessing the mechanisms of therapeutic benefits is needed because they are directly important for optimising scaffold physicochemical properties to optimally mediate external stimulation. The safety of nanomaterials requires confirmation by studies in humans and other animals in vivo, as this is crucial to the essential transformation to clinical applications. Although much work remains, we believe that rationally designed nanomaterials represent promising biofabrication strategies that will overcome most of the obstacles associated with repairing damaged bone.

## Figures and Tables

**Figure 1 nanomaterials-13-01449-f001:**
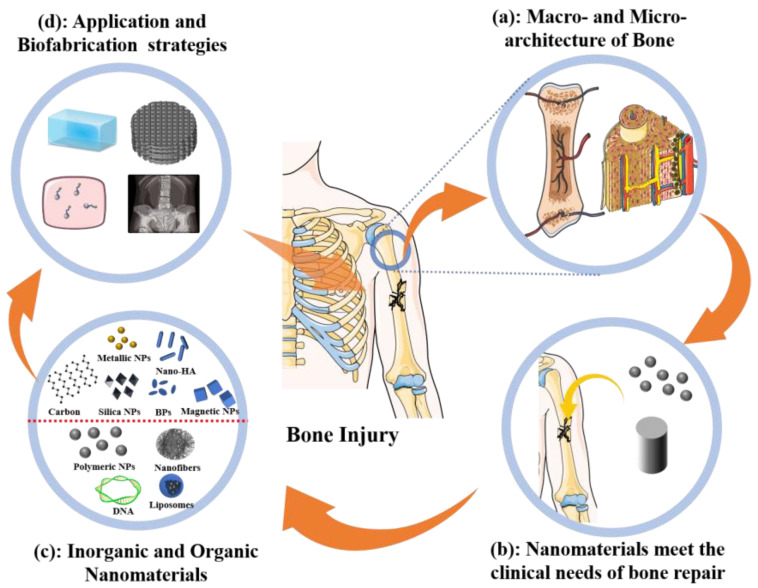
Scope of this review. Hierarchical architecture of natural bone from macro to nano will guide biofabrication strategies of bone engineering materials (**a**). Nanomaterials are promising for bone regeneration (**b**) and comprise inorganic and organic types (**c**). Applications of nanomaterials to engineered bone tissues (**d**).

**Figure 2 nanomaterials-13-01449-f002:**
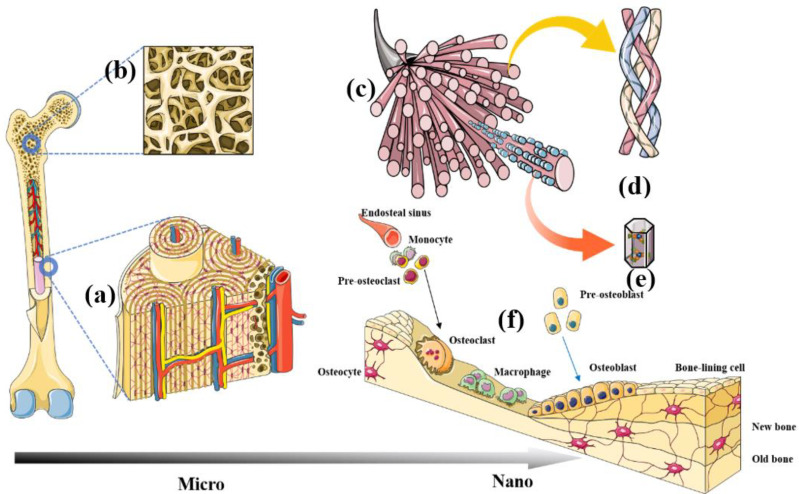
Schema of hierarchical architecture of bone tissue from macro- to nanoscales. Cortical bone and Haversian systems (**a**). Needle-shaped or irregular rod-shaped trabecular bone (**b**). Bone biomineral matrix (**c**). Collagen fibre (**d**). Nano-hydroxyapatite (**e**). Bone cells (**f**).

**Figure 3 nanomaterials-13-01449-f003:**
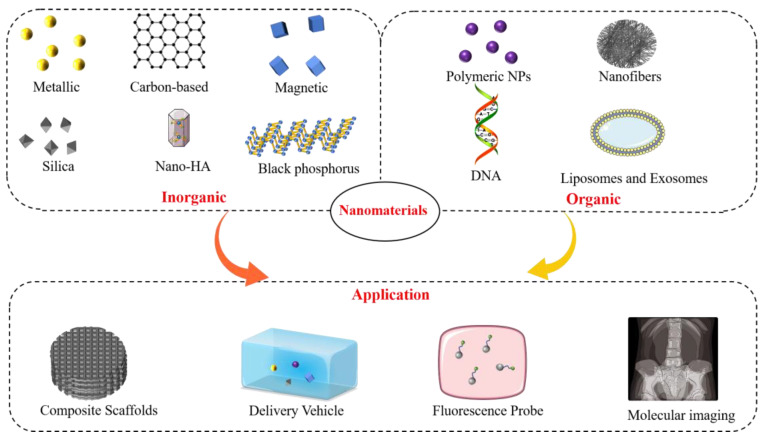
Classification and main applications of nanomaterials in bone tissue engineering.

**Figure 4 nanomaterials-13-01449-f004:**
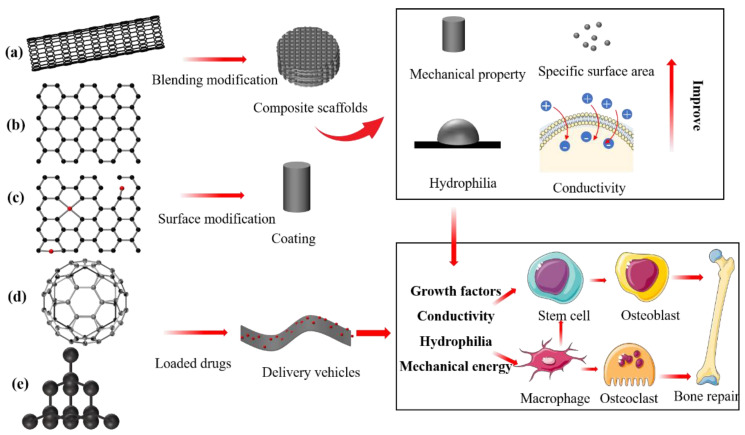
Schema of molecular structure of carbon-based family nanomaterials. (**a**) Carbon nanotubes. (**b**) Graphene. (**c**) Reduced graphene oxide. (**d**) Fullerenes. (**e**) Nanodiamonds. These carbon-based nanomaterials can be prepared into bone-repair scaffolds with various properties or loaded with bioactive factors and directly (growth factors) or indirectly (conductive or hydrophilically) promote bone tissue regeneration.

**Figure 5 nanomaterials-13-01449-f005:**
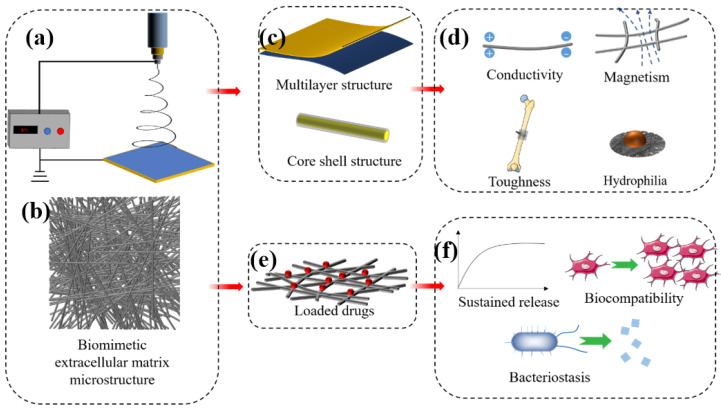
Diagram of nanofibre morphology (**b**) and preparation (**a**). Nanofibres can mimic the fibrous structure of natural extracellular matrix (**b**) and can be prepared into multilayer or core-shell structure (**c**) to endow it with specific physical and chemical properties (**d**). As a delivery vehicle (**e**), it can achieve sustained release of drugs, promote cell proliferation, or inhibit bacteria (**f**).

**Figure 6 nanomaterials-13-01449-f006:**
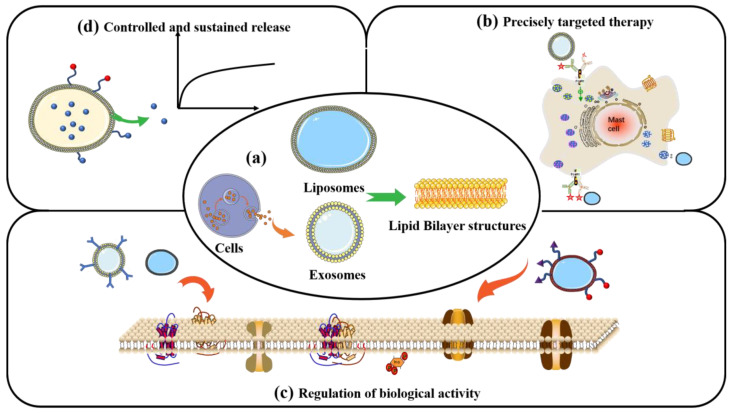
Liposomes and exosomes have similar lipid bilayer nanostructures (**a**). Liposomes and exosomes designed to store active targeting molecules (**b**), regulate intracellular cell bioactivities associated with bone regeneration (**c**), and release drugs (**d**) under regulated conditions.

**Table 1 nanomaterials-13-01449-t001:** Common nanoclay types according to structure. T represents tetrahedral units; O represents octahedral units.

Structure		Nanoclay Types
Layered	T:O	Kaolinite, Halloysite, rectorite
	T:O:T	Pyrophyillite, Illite, Vermiculite, Chlorite, Smectite, Montmorillonite
Fibrous		Attapulgite

**Table 2 nanomaterials-13-01449-t002:** Unique characteristics and typical applications of nanomaterials in bone tissue engineering.

Nanomaterials			
Inorganic		Unique Characteristics	Typical Application
	Metallic NPs (Au, Ag)	1. Biocompatibility 2. Antimicrobial 3. Controllable shape and size 4. Conductivity 5. Easy to functionalise	1. Metallic NP-based composite 2. Biological imaging 3. Diagnostic nanoprobes 4. Treating infections 5. Drug delivery
	Carbon-based Nanomaterials (Graphene, CNTs)	1. Excellent conductivity 2. Reinforce mechanical performance 3. Hydrophilia 4. Large specific surface area 5. Chemical stability 6. Thermal and wear resistance	1. Conductive composite 2. Imaging 3. Drug delivery 4. High mechanical properties implant
	Silica NPs	1. Controlled mesoporous structure 2. Conjugate with wide variety of compounds 3. Large specific surface area 4. Promotes osteogenic differentiation	1. Drug delivery systems 2. Biological imaging 3. Composite scaffold
	Nano-HAp	1. Natural bone tissue components 2. Outstanding osteogenic features 3. Easy chemical grafting 4. Protein adsorption 5. Improve the mechanical properties	1. Bone biomimetic scaffold 2. Surface modification of the bone implant 3. Loaded with biological factors 4. Artificial bone materials
	Black Phosphorus (BP)	1. Conductivity and piezoelectricity 2. Degraded into nontoxic phosphates 3. Be composed of phosphorus, an inorganic component of bone 4. Near-infrared optical properties 5. Reinforce mechanical performance	1. Conductivity scaffolds 2. NIR light-triggered drug release system 3. Photothermal therapy
	Magnetic NPs (Fe_2_O_3_, Fe_3_O_4_)	1. Superparamagnetic property 2. Magnetothermal 3. Magnetic imaging 4. Regulating osteogenic activity by the inherent magnetic field	1. Magnetic bone-repair scaffold 2. Magnetic nanoimaging technology 3. Magnetothermal therapy
	Nanoclays (MMT, HNTs)	1. Enforce mechanical performance 2. Charge heterogeneity 3. Promote osteogenesis releasing Mg^2+^ and Ca^2+^ 4. Easy chemical grafting 5. Protein adsorption	1. Drug delivery 2. As Hard polymeric scaffold reinforcers 3. As hydrogel reinforcers
Organic			
	Polymeric NPs	1. Nontoxic degradation products 2. Functional modification 3. Biocompatible	1. Drug delivery 2. Fluorescence imaging
	Nanofibres	1. Biomimetic structure of the ECM 2. Good tensile mechanical properties 3. Pore interconnectivity 4. High physical adsorption capacity	1. Be applied as bone-repair scaffolds 2. Drug delivery 3. Core-shell structure endow with specific biofunction 4. Nanofibre-reinforced bone composite scaffold
	DNA Nanomaterials	1. Be accurately designed, modified, and endowed with unique functions 2. Be composed of nucleobases	1. Drug delivery 2. Targeted therapy 3. Biofunctional modification of bone-repair scaffold
	Liposomes, Exosomes	1. Lipid bilayer structure 2. Easy to functionalise 3. Exosomes are extracellular vesicles that are secreted by cells and carry biomolecules	1. Drug delivery 2. Targeted therapy 3. Biofunctional modification of bone-repair scaffold

## Data Availability

Not applicable.

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
