# Peer review of "Tissue-Engineered Nanomaterials Play Diverse Roles in Bone Injury Repair"

_nanomaterials, 2023, doi:10.3390/nano13091449_

Round 1

Reviewer 1 Report

Following a thorough examination of the review paper entitled "Tissue-engineered nanomaterials play diverse roles in bone injury repair", I consider it I can be published in the present form, after few minor adjustments. However, I have a few points to make, which I will address in the following:

1. On page 5, line 154, the sentence "They can differentiate into osteocytes...." is not correct because the osteocytes cannot differentiate into the same type of cells.

2. On page 6, line 210, I suggest to the authors to reformulate the title "Nano-materials" to make the content of this section more suggestive. According to Figure 1, I have noticed that this section covers both the aspects (c) and (d), regarding the purpose of the present work.

3. On page 10, line 414, the sentence "Radioactive tracers analysed and evaluated using MRI..." should be replaced with a correct form. I belive that the authors have misunderstood the quoted article, because radiotracers cannot be visualized by the MRI technique. In the cited article, the radiotracers were visualized by SPECT-CT.

Reviewer 2 Report

The review titled "Tissue-engineered nanomaterials play diverse roles in bone injury repair" offers an interesting overview of the use of nanomaterials in the tissue engineering of bone. In the opinion of the referee the review can be published without additional revisions.

Author Response

Thanks for your comment and your effort to review our work.

Reviewer 3 Report

The authors presented review on nanomaterials roles on bone tissue engineering application in the manuscript entitled "Tissue-engineered nanomaterials play diverse roles in bone injury repair." Because the provided work is lacking in many details, it should only be considered for publication after a thorough revision.

 1.     The abstract must be updated so that it corresponds to the description in the main text. It should correspond to the review's content. Examples include early diagnosis, targeted therapy, and theoretical and technological bases insufficiently addressed in the main text.

2.     In order to comprehend the scope of the investigation, Figure 1 should be appropriately labeled and described in the introduction section.

3.     On page 2, authors described “Nano-based materials for bone tissue engineering are needed for clinical applications.”  Why is this so crucial in clinical applications? Explain more details here.

4.     The description should follow a chronological structure. For example, Figure 2a should be explained before to Figure 2b, and Figure 2c should be reviewed prior to Figure 2d.

5.     “Four main types of bone cells function in the physiological process of fracture healing: osteoblasts, osteocytes, osteoclasts, and extracellular lining cells (Fig. 3f).” There is typo with Figure number.

6.     Mechanical and electrical properties are essential for bone tissue engineering, as described by the authors. But, they are not thoroughly discussed? The mechanical and electrical properties of the native tissue should be compared and addressed in the each section.

7.     A comparison table of nanomaterials and their properties should be provided.

8.     Nanoclays are also well studied for bone tissue engineering. A new section about nanoclays should be included.

9.     A thorough language correction should be performed

Reviewer 4 Report

The article "Tissue-engineered nanomaterials play diverse roles in bone injury repair" aims to describe the current state of the art in domain of biomaterials used in bone repair. It is a valuable study that can be published after authors address the following problems:

Abstract should be checked and revised carefully by briefly introducing the work plan and key findings. Abstracts should highlight the features of this review, as often abstract section is presented separately in search engines, it must be able to stand alone as an informative piece.

While the objective of this review is clear, authors should indicate also the review methodology (keywords used, databases consulted, time interval, other criteria). For review criteria please see PRISMA for example.

The English language needs some polishing for style and typos (e.g. “ovalently” row 475)

This work is interesting and can be boosted further. Following literature could prove this manuscript doi: 10.3390/ijms232416180; doi: 10.3390/nano9070985; doi: 10.3390/ijms21218082; doi: 10.1016/j.matlet.2018.08.042

Section 4.1.3 title is “silica nanoparticles”, SiO2, but in the body text authors mention multiple times silicon nanoparticles, Si – please correct it; same issue at row 307; By consulting the references mentioned by authors is clear that the subject is mesoporous silica (SiO2 nanoparticles, or nanostructured MCM 41/48 or SBA 15).

Black phosphorous is not discovered in 2014 (see https://doi.org/10.1038/s42254-019-0043-5). Maybe 1914?

Rows 112 and 473 declare different percent of collagen in human bone (90% vs 70-86%). What are your thoughts about it?

Reviewer 5 Report

Review article 'Tissue-engineered nanomaterials play diverse roles in bone injury repair' focuses on the use of nanomaterials in bone tissue engineering. It includes description and discussion of bone structure, classification of nanomaterials and potential of their use in bone repair. The review is relatively small and superficial, with short bibliography (106 references). However, the most of references are actual, the review theme is of interest for a wide range of scientists and physicians, and the article can be published in Nanomaterials journal.

Comments:

1. Line 32 and below: please insert spaces between words and reference numbers in square brackets.

2. Lines 124-130: is it accurate to discuss here (very superficially) HAp-based composites?

3. Line 323 and below: strictly speaking, 'pure' HAp of the formula Ca10(PO4)6(OH)2 is not a bone apatite that contains carbonate and sodium ions [Materials 11 (2018) 1813; J. Solid State Chem. 255 (2017) 27; Minerals 12 (2022) 170]. HAp has reduced biodegradability in comparison with other Ca-P phases [J. Ceramic Soc. Japan 127 (2019) 595]. HAp grafting for compatibilization with polymer matrix is only mentioned, a number of recent interesting works are missed [Compos. A Appl. Sci. Manuf. 132 (2020) 105841; Coll. Polym. Sci. 299 (2021) 623; Coll. Surf. B Biointerfaces 217 (2022) 112668; Mater. Design 201 (2021) 109490; Compos. A Appl. Sci. Manuf. 162 (2022) 107130;  Nanomaterials 11 (2021) 249; Nanomaterials 11 (2021) 213; J. Mater. Chem. B 10 (2022) 214; Int. J. Mol. Sci. 22 (2021) 7690]. I recommend to rename the section (nano-HAp and other bone mineral substitutes) and add some references on TCP, CAp and other prospective phases. Also note that the nature and size of macrophage polarization etc. depend on the size and nature of bone mineral substitute.

4. Line 418 and below: the problem of the acidic inflammatory response when using biodegradable polyesters should be mentioned here, with appropriate references [Acta Biomater. 65 (2018) 259].

Round 2

Reviewer 3 Report

After extensive revision, the review article has been considerably improved by the authors. Therefore , I recommend that the manuscript be considered for publication in Nanomaterials.

Reviewer 4 Report

The authors have responded to my comments and have addressed all my concerns, substantially improving the manuscript, therefore, I suggest publishing the paper in the current form.